# Diabetes and the Prostate: Elevated Fasting Glucose, Insulin Resistance and Higher Levels of Adrenal Steroids in Prostate Cancer

**DOI:** 10.3390/jcm11226762

**Published:** 2022-11-15

**Authors:** Stefan Zoltán Lutz, Jörg Hennenlotter, Andras Franko, Corinna Dannecker, Louise Fritsche, Konstantinos Kantartzis, Róbert Wagner, Andreas Peter, Norbert Stefan, Andreas Fritsche, Tilman Todenhöfer, Arnulf Stenzl, Hans-Ulrich Häring, Martin Heni

**Affiliations:** 1Institute for Diabetes Research and Metabolic Diseases (IDM) of the Helmholtz Center Munich, University of Tübingen, 72074 Tübingen, Germany; 2German Center for Diabetes Research (DZD), 85764 Munich, Germany; 3Clinic for Geriatric and Orthopedic Rehabilitation Bad Sebastiansweiler, 72116 Mössingen, Germany; 4Department of Urology, University of Tübingen, 72076 Tübingen, Germany; 5Department of Internal Medicine, Division of Endocrinology, Diabetology and Nephrology, University of Tübingen, 72074 Tübingen, Germany; 6Department for Diagnostic Laboratory Medicine, Institute for Clinical Chemistry and Pathobiochemistry, University Hospital of Tübingen, 72076 Tübingen, Germany; 7Department of Internal Medicine 1, Division of Endocrinology and Diabetology, University Hospital Ulm, 89081 Ulm, Germany

**Keywords:** insulin resistance, hyperglycemia, prostate cancer

## Abstract

Although epidemiological studies suggest a lower prostate cancer incidence rate in patients with type 2 diabetes, cancer survival is markedly reduced. Underlying mechanisms that connect the two diseases are still unclear. Potential links between type 2 diabetes and prostate cancer are hallmarks of the metabolic syndrome, such as hyperglycemia and dyslipidemia. Therefore, we explored the systemic metabolism of 103 prostate cancer patients with newly diagnosed and yet untreated prostate cancer compared to 107 healthy controls, who were carefully matched for age and BMI. Here, we report that patients with prostate cancer display higher fasting blood glucose levels and insulin resistance, without changes in insulin secretion. With respect to lipid metabolism, serum triglyceride levels were lower in patients with prostate cancer. In addition, we report increased adrenal steroid biosynthesis in these patients. Our results indicate that higher fasting glucose levels in patients with prostate cancer may be explained at least in part by insulin resistance, due to the enhanced synthesis of adrenal steroids.

## 1. Introduction

Epidemiological studies suggest a lower prostate cancer (PCa) incidence in patients with type 2 diabetes [1,2]. However, the cancer survival in the case of coexisting diabetes is clearly reduced [3]. Both diabetes and PCa rank among the most frequent diseases in males worldwide with an enormous impact on morbidity and mortality. Although both diseases share several common risk factors, the precise causal links between them are not fully understood.

Important factors that may link PCa and diabetes are hallmarks of the metabolic syndrome, i.e., hyperglycemia and dyslipidemia [4]. In general, patients with higher blood glucose levels have more aggressive PCa [5]. Strong evidence exists for the impact of hyperglycemia on PCa recurrence after primary surgical or radiation therapy [6] and hyperglycemia-induced chemoresistance of PCa [7]. Vice versa, the commonly used first line androgen deprivation therapy for advanced PCa often causes metabolic disturbances, namely insulin resistance and hyperglycemia [8].

Recently, we could show a markedly higher prevalence of lymph node metastasis and more high-grade tumors in prostate cancer patients with coexistent diabetes [9]. One possible explanation for this phenomenon could be the elevated androgen receptor signaling in men with diabetes [10]. Besides these alterations in prostatic tissue, the systemic metabolism could contribute to it. Therefore, we investigated the metabolic profile of PCa patients in comparison to well-matched healthy controls.

## 2. Material and Methods

For this purpose, we phenotyped 103 patients with newly diagnosed and yet untreated PCa who were recruited in the Department of Urology, University of Tübingen prior to radical prostatectomy (for tumor characteristics, see Appendix A), and 107 male controls without PCa from the Tübingen Family Study [11] who were carefully matched for age and BMI. Patients with a history of malignancy other than prostate were excluded. To avoid interference by glucose-lowering medication, we excluded patients with overt diabetes. The insulin sensitivity index was calculated as proposed by Matsuda and deFronzo based on a 5-point oral glucose tolerance test. Insulin secretion was calculated as the area under the curve (AUC) according to the trapezoid method as ½[½(C-Pep_0′_) + C-Pep_30′_ + C-Pep_60′_ + C-Pep_90′_ + ½(C-Pep_120_)]/½[½(Glc_0′_) + Glc_30′_ + Glc_60′_ + Glc_90′_ + ½(Glc_120′_)], with C-Pep = C-peptide. Insulin secretion was analyzed after adjustment for insulin sensitivity [12].

Liver fat was assessed by ^1^H-MR spectroscopy in a subset of 114 subjects. The Ethics Committee of the University of Tübingen approved the protocol and all participants provided written informed consent.

Plasma concentrations of glucose, wide range C-reactive protein, triglycerides (TGs) and total-, HDL- and low-density lipoprotein-cholesterol were measured using the ADVIA XPT clinical chemical analyzer (Siemens Healthineers, Eschborn, Germany). Plasma concentrations of total non-esterified fatty acid (NEFA) were measured with an enzymatic method (WAKO Chemicals, Neuss, Germany) on the latter instrument.

Plasma insulin, C-peptide, estradiol, cortisol and progesterone were determined using the ADVIA Centaur XP Immunoassay System, and DHEAS as well as androstenedione were measured using the Immulite 2000 XPi system (both Siemens Healthineers, Eschborn, Germany).

The matching of patients with PCa to similar participants without PCa was performed using a nearest match procedure with the package “MatchIt” in R (http://www.jstatsoft.org/v42/i08/, accessed on 12 January 2019). Statistical analyses were conducted using JMP 14.0 (SAS Institute Inc., Cary, NC, USA). Associations of PCa status with basal glucose and insulin sensitivity were tested in multivariate linear regression analyses of log-transformed data with adjustment for age and BMI, while associations with insulin secretion were tested after adjustment for age and insulin sensitivity. A *p*-value < 0.05 was considered to be statistically significant.

## 3. Results

Compared to the controls, here we report that PCa patients displayed higher fasting blood glucose levels (5.65 ± 0.05 mmol/L and 5.44 ± 0.05 mmol/L, respectively, *p* = 0.002), independently of age and BMI. In parallel, this went along with reduced insulin sensitivity in the PCa group (9.1 ± 0.5 AU and 10.3 ± 0.5 AU, respectively, *p* = 0.016, Figure 1). By contrast, patients with PCa displayed no significant difference in insulin secretion compared to controls (*p* = 0.66, Table 1). The PSA-levels in patients with PCa were 9.27 ± 0.89 ng/mL with a median of 6.9 [0.47–59]. PSA levels in the control group were not available. Regarding lipid metabolism, serum triglyceride levels were lower in patients with PCa (95.4 ± 5.9 mg/dL and 130.6 ± 5.8 mg/dL, respectively, *p* < 0.0001), while no differences between the two groups were detectable in NEFA and cholesterol levels (*p* = 0.35 and *p* = 0.06, respectively, Table 1). Moreover, our results indicate increased steroid synthesis in PCa patients, as they displayed elevated serum cortisol (476.1 ± 14 nmol/L and 438 ± 13.9 nmol/L, respectively, *p* = 0.029), androstenedione (6.5 ± 0.3 nmol/L and 4.8 ± 0.3 nmol/L, respectively, *p* < 0.0001), progesterone (1.3 ± 0.04 nmol/L and 0.8 ± 0.04 nmol/L, respectively, *p* < 0.0001), DHEA-Sulfate (4.2 ± 0.2 µmol/L and 2.9 ± 0.2 µmol/L, respectively, *p* < 0.0001) and estradiol (126.4 ± 3.1 pmol/L and 115.5 ± 3.0 pmol/L, respectively, *p* = 0.009) levels (Figure 2).

## 4. Discussion

Although epidemiological studies indicated lower PCa risk in men with type 2 diabetes, the precise relationships between PCa and hyperglycemia are not well understood. In particular, it is unclear whether hyperglycemia may modulate PCa risk and vice versa. Moreover, antidiabetic medication may also impact on PCa risk through different mechanisms independently of lowering blood glucose levels [13]. In the Finnish Randomized Study of Screening for Prostate Cancer (FinRSPC) with a median follow-up of 14.7 years, increased fasting glucose was identified as a predictor of PCa development [13]. Although we excluded patients with diabetes, men with PCa in our study clearly displayed higher fasting glucose levels, compared to carefully matched controls. Thus, our results are in line with the aforementioned Finnish study.

In another study, Gerlini et al. could show that glucose intolerance and insulin resistance regulate adipocyte transcription in human obesity [14]. As we found hyperglycemia and insulin resistance in PCa, it is tempting to speculate that comparable transcriptional effects might exist in the prostate.

Interestingly, the metabolic pattern of hyperglycemia in the context of low circulating triglycerides was associated with poor prognosis and increased risk of PCa death in a previous study [4]. The role of triglycerides in PCa is not fully understood yet. However, lipid metabolism seems essential for prostate cancer cells, while they do not depend on increased aerobic glycolysis [15]. As we detected lower triglycerides in PCa patients, while liver fat content was comparable to healthy controls, triglyceride uptake might be higher in prostate cancer cells. This hypothesis needs to be addressed in future studies.

One known determinant of insulin sensitivity/insulin resistance, which might further support our findings, are steroid hormones. PCa was reported to release miRNA, which affect steroid biosynthesis mainly in the adrenal gland [16,17]. We therefore measured key steroid hormones in our patients.

PCa is a hormone-dependent cancer, and androgen signaling represents a key driver for cancer cell growth. However, in a collaborative analysis investigating direct correlations between risk of PCa and serum concentrations of circulating sex steroids, no associations were found [18]. In spite of this, complex interactions between diabetes and sex steroids exist [19]. We hypothesized that hyperglycemia in PCa patients may be linked to the elevated synthesis of sex steroids. Testosterone synthesis has been reported to be reduced in patients with diabetes [20], but continued androgen receptor (AR) activation in the presence of low circulating levels of testosterone have been proposed, indicating alternative AR ligands to bypass the testosterone/dihydrotestosterone-mediated AR activation [10]. Therefore, we assessed adrenal steroids as androgen precursors. Indeed, we found elevated levels of cortisol, androstenedione, DHEA-Sulfate, progesterone and estradiol in PCa. Of note, serum levels of gonadal steroid testosterone were comparable between the groups (Table 1). The adrenal hormones may, on the one hand, activate at least mutated AR, a frequent mechanism in PCa to provoke resistance to hormonal therapy [21], but on the other hand, also elevate fasting glucose levels in PCa patients by introducing insulin resistance. Unfortunately, unlike in our study, most previous studies did not perform the rigorous matching for anthropometric determinants of steroid hormones and may, therefore, have missed to detect differentially regulated hormones in cancer vs. control. Nevertheless, our results are supported by a previous report showing higher levels of estradiol in PCa [22]. Several reports exist suggesting a proliferative role of estradiol in PCa especially via activating the estrogen receptor α [23]. Androstenedione seems relevant for AR activation with regard to being largely unaffected by ADT or orchiectomy in spite of testosterone/dihydrotestosterone [24].

Among the limitations of our work is the cross-sectional study design that can never fully uncover changes over time. The prostate cancer cases were heterogeneous in grade and stage which might have impacted our findings (see Appendix A). Due to the available sample size, we could not investigate potential differences due to cancer grade/stage in further subgroups in more detail. Clearly, additional experimental studies are needed to unravel the molecular links between steroid hormones, diabetes and PCa.

Our results suggest that higher fasting glucose levels in patients with PCa may be explained at least in part by insulin resistance, due to the enhanced synthesis of adrenal steroids. This may pave the way for new therapeutic approaches in coexisting diabetes and PCa.

## Figures and Tables

**Figure 1 jcm-11-06762-f001:**
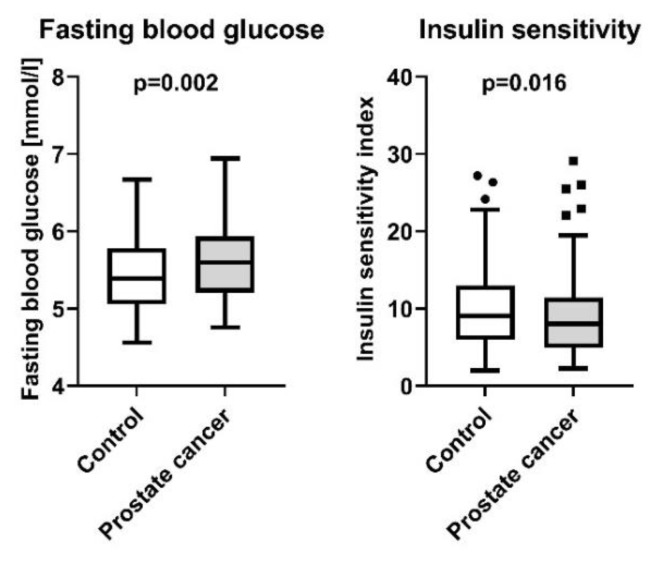
Fasting blood glucose and insulin sensitivity index in 103 patients with newly diagnosed and yet untreated PCa, and 107 male controls without PCa from the Tübingen Family Study. Presented are box plots with whiskers indicating 1.5 interquartile range.

**Figure 2 jcm-11-06762-f002:**
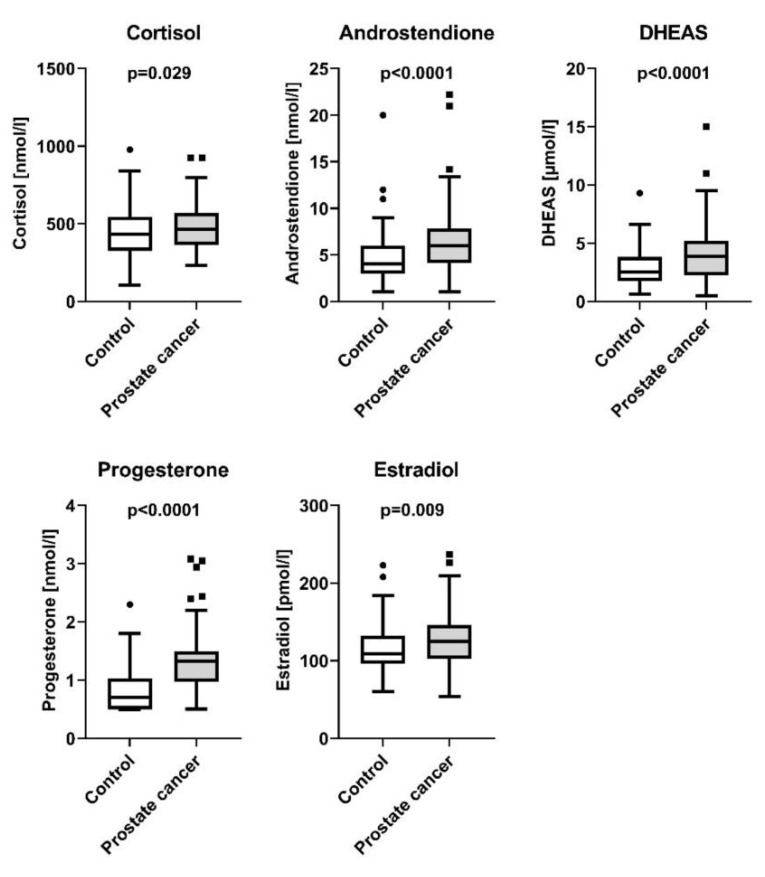
Steroid hormones in controls and patients with prostate cancer. Presented are box plots with whiskers indicating 1.5 interquartile range.

**Table 1 jcm-11-06762-t001:** Characteristics of the study population. Included are 103 patients with newly diagnosed and yet untreated PCa, and 107 male controls without PCa from the Tübingen Family Study. Abbreviations: AUC, area under the curve; BMI, body mass index. Data are given as mean ± SEM for all parameters. *p*-values are given for unadjusted data, *p**-values are for a model adjusting for age and BMI, except the associations with insulin secretion which are adjusted for age and insulin sensitivity.

	Control		Prostate Cancer			
	Mean	SEM	Mean	SEM	*p*	*p**
**Age (y)**	63.68	0.77	63.47	0.78	0.94	-
**BMI (kg/m^2^)**	26.88	0.33	26.73	0.34	0.76	-
**Insulin, fasting (pmol/L)**	78.09	4.8	90.84	4.89	0.004	0.0004
**C-peptide, fasting (pmol/L)**	529.07	22.17	552.5	22.6	0.28	0.13
**C-peptide, 120 min (pmol/L)**	2528.69	102.23	2366.45	103.71	0.41	0.45
**Non-esterified fatty acid (µmol/L)**	579.1	25.32	566.44	25.57	0.34	0.35
**Triglyceride (mg/dL)**	130.59	5.79	95.41	5.93	<0.0001	<0.0001
**Cholesterol (mg/dL)**	205.55	3.68	195.36	3.97	0.07	0.06
**HDL-cholesterol (md/dL)**	51.72	1.14	52.41	1.23	0.74	0.84
**LDL-cholesterol (ml/dL)**	123.32	3.29	118.36	3.55	0.37	0.36
**C-reactive protein (mg/dL)**	0.14	0.1	0.32	0.14	0.45	0.36
**Intrahepatic lipids (%)**	6.14	0.88	5.95	0.72	0.15	0.21
**AUC C-peptide 0–120/AUC glucose 0–120**	261.97	8.05	267.75	8.25	0.42	0.66
**AUC C-peptide 0–30/AUC glucose 0–30**	153.11	5.39	156.67	5.49	0.47	0.47
**Testosterone (nmol/L)**	13.06	0.46	13.28	0.46	0.98	0.9

## Data Availability

The data are not publicly available as they contain information that could compromise research participant privacy/consent.

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
