# Peer review of "Diabetes and the Prostate: Elevated Fasting Glucose, Insulin Resistance and Higher Levels of Adrenal Steroids in Prostate Cancer"

_jcm, 2022, doi:10.3390/jcm11226762_

Round 1

Reviewer 1 Report

Aim of this brief report is to compare metabolic and hormonal parameters of a cohort of patients with untreated prostate cancer and male controls without PCa. The aim is clear and meaningful and the potential clinical implications from a diagnostic and therapeutic point of view could be relevant.

The manuscript is well-written. References are appropriate.

I have some minor comments:

- Tumor features of patients with prostate cancer (including median PSA levels -this feature should reported also for patients without PCa if available-, Gleason Score groups and stage at diagnosis) should be reported.

- Limitations of this study (including study design and limited sample size) should clearly reported in the last of part of conclusion section of the manuscript.

-Title should be rewritten to make it more informative and catchy. 

Author Response

We thank the reviewer for his/her helpful comments.
They have helped to significantly improve the quality of our manuscript. Here are our answers point by point.

Tumor features of patients with prostate cancer (including median PSA levels
-this feature should reported also for patients without PCa if available-,
Gleason Score groups and stage at diagnosis) should be reported.

Thank you for this comment. Now we added tumor features of all 103 patients with prostate cancer including median PSA levels, Gleason score groups und stages at diagnosis as indicated in the following: PSA levels of the PCa group are given in the results section lines 93-94. PSA levels for patients without PCa are unfortunately not available. This is also indicated in the results section.
Gleason scores and stages at diagnosis are given in the new Supplementary data Table 1 and Table 2.

- Limitations of this study (including study design and limited sample size)
 should clearly reported in the last of part of conclusion section of the
manuscript.

We apologize for not having this in the initial version. We have now added a limitations paragraph to the Discussion. 

Title should be rewritten to make it more informative and catchy.

Thank you for this comment. We have rewritten the title as indicated in the
uploaded version of the manuscript.  

Reviewer 2 Report

The authors report a cross-sectional comparison of several metabolic and hormone markers among prostate cancer cases and controls. 

The terminology “whole body metabolism” is odd.  Perhaps metabolic profile would be more appropriate?

The authors need to report more information on how cases and controls were selected and the overall data collection elements.  Was blood collected after fasting?  What is meant by “overt diabetes”?  Is there a reference to the Tübingen Family Study that could be included? How heterogeneous are the prostate cancer cases with respect to grade/stage etc.?  If some men have more advanced disease, this could substantially impact their levels of metabolic/hormone markers.

In the results, line 97, the authors refer to “adrenal steroid synthesis”, though some hormones evaluated are gonadal in origin (e.g. progesterone, estradiol, testosterone).

Throughout results, it would be helpful to refer to the comparison group of "controls" when mentioning significant results.

Figures.  Bar graphs, which are primarily used for count data, are not adequate to visualize these data.  I would suggest either putting these data in a table or a using a different type of figure (such as a box plot or even just plotting the mean with error bars) to visually show the mean and SE.

Table 1. The title “Characteristics of the study population” is not very accurate here.  Were any other participant characteristics (race, education, etc). that could be reported?  While these factors may not be substantially relevant to the metabolic markers evaluated, they would allow readers to determine the comparability of cases and controls.

 Table 1. There are no real differences between the adjusted and unadjusted.  It may be better to simply report the adjusted.  The authors should explain/discuss their choice of adjusting insulin secretion for insulin sensitivity, as this analysis seems to be testing a different question.

Conclusions.  This section should be re-titled as the Discussion.  

The discussion is  lacking in comparison to previous studies that have evaluated fasting glucose/insulin/markers of insulin resistance/steroid hormones and PCa.

The authors should at least mention the potential limitations of their study, including those related to study design.  It would be helpful for the authors to interpret their data within the context of the cross-sectional study design they used.

In the final conclusion paragraph, the statement “Our results indicate that higher fasting glucose levels in patients with PCa may be explained at least in part by insulin resistance, due to enhanced synthesis of adrenal steroids.”  does not appear to be substantiated by the analyses presented.  This statement remains speculative given the analyses presented by the authors.

Author Response

We thank the reviewer for his/her helpful comments.
They have helped to significantly improve the quality of our manuscript. Here are our answers point by point.

The terminology “whole body metabolism” is odd.  Perhaps metabolic profile would be more appropriate?

Thank you for this comment. We changed the wording as suggested on page 2 line 54. 

The authors need to report more information on how cases and controls were selected and the overall data collection elements.  Was blood collected after fasting?  What is meant by “overt diabetes”?  Is there a reference to the Tübingen Family Study that could be included? How heterogeneous are the prostate cancer cases with respect to grade/stage etc.?  If some men have more advanced disease, this could substantially impact their levels of metabolic/hormone markers.

Thank you very much for these important comments. We added more information to the selection of cases and indicated the reference to the Tübingen Family Study as required. Blood was collected after overnight fasting. With the terminus "overt diabetes" is a state of already diagnosed diabetes mellitus defined, where criteria of diabetes are fulfilled. Thank you very much also for the comment on tumor heterogeneousity. For this we provide additional data in the new Supplementary material Table 1 und Table 2, where we indicate Gleason-Scores and tumor stages of the PCa patients. We also added this potential weakness to the limitations part of the manuscript.  

In the results, line 97, the authors refer to “adrenal steroid synthesis”, though some hormones evaluated are gonadal in origin (e.g. progesterone, estradiol, testosterone).

Thank you for this comment. We deleted the word "adrenal" in line 97. 

Throughout results, it would be helpful to refer to the comparison group of "controls" when mentioning significant results.

We changed wording in the results section to "controls". 

Figures.  Bar graphs, which are primarily used for count data, are not adequate to visualize these data.  I would suggest either putting these data in a table or a using a different type of figure (such as a box plot or even just plotting the mean with error bars) to visually show the mean and SE.

Thank you for this helpful comment. We changed the presentation of the data in the two figures by using box plots as recommended. 

Table 1. The title “Characteristics of the study population” is not very accurate here.  Were any other participant characteristics (race, education, etc). that could be reported?  While these factors may not be substantially relevant to the metabolic markers evaluated, they would allow readers to determine the comparability of cases and controls.

Thank you. All participants of the study were caucasians. Unfortunately we have no data regarding education, that could be reported. From the OGTTs and from the clinical characterization, there is many more data available. However, we do not feel that adding more additional metabolic parameters would help in the interpretation of the results. If the reviewer or the editor feels that specific information would be helpful, we are of course ready to put it in. 

Table 1. There are no real differences between the adjusted and unadjusted.  It may be better to simply report the adjusted.  The authors should explain/discuss their choice of adjusting insulin secretion for insulin sensitivity, as this analysis seems to be testing a different question.

We discussed this with our statistical expert. He recommended to present both unadjusted and adjusted p-values to make clear that the results are independent of the tested confounders. When analyzing insulin secretion, this should always be done after adjustment for insulin sensitivity, as insulin resistance is one of the strongest determinants of insulin secretion (see e.g. Ahrén et al. Eur J Endocrinol. 2004). We have added this information to the Methods part of the manuscript. 

Conclusions.  This section should be re-titled as the Discussion.  

We re-titled the last section as requested. 

The discussion is  lacking in comparison to previous studies that have evaluated fasting glucose/insulin/markers of insulin resistance/steroid hormones and PCa.

Previous epidemiological studies reported that in men, pharmacological lowering of circulating testosterone levels by androgen deprivation therapy for the treatment of prostate cancer increases diabetes risk. Although numerous studies suggest an association between testosterone and either insulin sensitivity or insulin secretion, no systematic investigations of these factors with glucose metabolism in a large group in prostate cancer patients have been published so far.

The authors should at least mention the potential limitations of their study, including those related to study design.  It would be helpful for the authors to interpret their data within the context of the cross-sectional study design they used.

We apologize for not having this in the initial version of the manuscript. We have now added a limitations paragraph. 

In the final conclusion paragraph, the statement “Our results indicate that higher fasting glucose levels in patients with PCa may be explained at least in part by insulin resistance, due to enhanced synthesis of adrenal steroids.”  does not appear to be substantiated by the analyses presented.  This statement remains speculative given the analyses presented by the authors.

We changed the wording as indicated, and replaced the word "indicate" with "suggest". 

Round 2

Reviewer 2 Report

Thank you for addressing the majority of my comments.  For figures 1 & 2, please make sure to indicate what data the figures are showing (ie means+/- SEM's) - mention of this has been deleted.

Author Response

Thank you for notifying us of this missing information that was accidently removed in the first round of revision. We now define the presented box plots in the figure legends.